# Efficacy of Over-The-Scope Clip Method as a Novel Hemostatic Therapy for Colonic Diverticular Bleeding

**DOI:** 10.3390/jcm10132891

**Published:** 2021-06-29

**Authors:** Koichiro Kawano, Mamoru Takenaka, Reiko Kawano, Daisuke Kagoshige, Yuta Kawase, Tomonori Moriguchi, Hiroshi Tanabe, Takao Katoh, Katsuhisa Nishi, Masatoshi Kudo

**Affiliations:** 1Departments of Gastroenterology, Hyogo Prefectural Awaji Medical Center, Sumoto 656-0021, Japan; toranoana18@hotmail.com (K.K.); reikooooo4@gmail.com (R.K.); kagoj.macan@gmail.com (D.K.); y.kawase0222@gmail.com (Y.K.); tomonori_goodlife@hotmail.com (T.M.); hakuhoku0922tanabe@gmail.com (H.T.); takao09041026550@yahoo.co.jp (T.K.); katsuhisa@awajimc.jp (K.N.); 2Departments of Gastroenterology and Hepatology, Kindai University Faculty of Medicine, Osaka-Sayama 589-8511, Japan; m-kudo@med.kindai.ac.jp

**Keywords:** OTSC, colonic diverticular bleeding, endoscopic hemostasis

## Abstract

Colonic diverticular could bleed recurrently, and, sometimes, fatal massive bleeding could occur. However, the choice of endoscopic hemostasis remains controversial. Although the over-the-scope clip (OTSC) method has been reported to be effective, it has not been fully evaluated due to the small number of cases. This study aimed to evaluate the efficacy of the OTSC method for colonic diverticular bleeding. Between August 2017 and December 2020, 36 consecutive patients, including those who could not be treated using endoscopic band ligation (EBL) and those in whom re-bleeding had occurred after EBL, underwent the OTSC method for hemostasis of colonic diverticular bleeding at Hyogo Prefectural Awaji Medical Center. The procedure success rate, adverse events rate, early phase re-bleeding rate (within 30 days following primary hemostasis), and the requirement rate for additional transcatheter arterial embolization (TAE) or surgery were the outcomes assessed. The outcomes were procedure success rate 100%, adverse events rate 0%, early phase re-bleeding rate 8.3%, and additional TAE or surgery rate 0%. These results suggest that the OTSC method is a safe and effective treatment for managing colonic diverticular bleeding.

## 1. Introduction

Colonic diverticular bleeding results in spontaneous hemostasis in 66–93% of cases [1,2,3], but could sometimes cause recurrent bleeding or fatal massive bleeding. Thus, when the source of bleeding can be identified, endoscopic hemostasis is the first choice of treatment. Nevertheless, no randomized controlled trial has evaluated respective treatments for endoscopic hemostasis in colonic diverticular bleeding, and the most effective endoscopic hemostatic method has not been ascertained. The over-the-scope clip (OTSC) (Ovesco Endoscopy AG, Tübingen, Germany) is a shape-memorized clip made of nitinol (Figure 1) that is used for suturing tissue or for securing hemostasis during gastrointestinal tract bleeding. Although there are some reports on the use of the OTSC method in the management of colonic diverticular bleeding, the studies are limited and involve few cases. Therefore, a case-accumulated study is necessary to establish evidence of the safety and efficacy of the OTSC method for colonic diverticular bleeding. Here, we performed a retrospective study based on a case series of hemostasis treated by the OTSC method for colonic diverticular bleeding.

## 2. Patients and Methods

A total of 115 patients underwent colonoscopy because of suspected acute lower gastrointestinal bleeding, and out of these, 36 consecutive cases that were diagnosed with colonic diverticular bleeding accompanied by stigmata of recent hemorrhage (SRH) [4] underwent endoscopic hemostasis using OTSC. SRH was defined as (1) active bleeding from a diverticulum; (2) non-bleeding visible vessels in a diverticulum; or (3) the presence of indication (1) or (2) when the adherent clot was removed [4].

### 2.1. The Over-the-Scope Clip (OTSC) Method

Colonoscopy was performed after hemodynamic stabilization by fluid infusion or blood transfusion if unstable. We used a colonoscope PCF290ZAI (Olympus, Tokyo, Japan) with a forward water supply function. An OTSC type T (diameter: 10 mm) was used. The process of hemostasis by OTSC is performed as follows: (1) place two marking clips around the bleeding diverticulum using an EZ Clip (Olympus, Tokyo, Japan), and then withdraw the colonoscope (Figure 2a). (2) After mounting the OTSC system to the tip of the scope, re-insert the colonoscope to the site of the bleeding diverticulum marked by the clip. (3) Release the OTSC over the diverticulum by rotating the hand wheel attached to the endoscopic forceps hole when the diverticulum is adequately towed by suction. If the diverticulum cannot be sufficiently inverted, the OTSC can be released with continuous suction (assistant forceps were not necessary in our cases). Before suction, it is necessary to adjust the position of the endoscopic tip mounted with the OTSC to the normal mucosa around the diverticulum, not to the inner wall of the diverticulum lacking mucosal layer. (4) Ensure that hemostasis has been achieved by suturing the diverticulum with the OTSC placement (Figure 2b).

### 2.2. Outcome Definitions

The outcomes evaluated were procedure success, adverse events, early phase re-bleeding, and additional TAE or surgery. Procedure success was defined as OTSC placement and cessation of active bleeding during endoscopic treatment, and early phase re-bleeding was defined as re-bleeding within 30 days following primary hemostasis.

### 2.3. Statistical Analysis

The study was a retrospective observational study, and EZR version1.54 was used for all statistical analyses. EZR is a statistical software that extends the functions of R and R Commander, and is distributed free of charge on the website of Saitama Medical Center attached to Jichi Medical University [5].

## 3. Results

### 3.1. Baseline Characteristics of Patients

The median age of the participants was 78 (range: 61–97) years, and 25 (69%) were males. Two patients (6%) were treated with endoscopic band ligation (EBL) before being treated with the OTSC method. In one of the two cases, hemostasis could not be achieved by EBL because suction of the responsible diverticulum was difficult, and the OTSC method was performed on the same day. In the other case, re-bleeding occurred three days after EBL owing to drop out of the band, and the OTSC method was performed. An antithrombotic drug was used in 14 patients (39%), steroid in five (14%), and NSAID in six (17%). An oral digestive tract cleaning agent was prepared in 19 patients (42%). Active bleeding was detected in 21 cases (58%), and visible vessels were observed in 15 cases (42%). The shock index on admission was 0.68 ± 0.18 (Table 1).

### 3.2. Treatment Outcome of OTSC Method

In all 36 cases (100%, 95% CI: 92–100%), we achieved procedure success and primary hemostasis. Early phase re-bleeding occurred in three cases (8.3%, 95% CI: 8–22.5%), and colonoscopy performed at the time of re-bleeding revealed the maintained state of inverted and bulging tissue at the site of the bleeding diverticulum that was treated with OTSC (Figure 3). As active bleeding was observed at the opening of the bleeding diverticulum in two of the three cases, re-hemostasis was achieved by placement of an OTSC at a shallower position than the previously placed OTSC. Fresh blood was pooled in the colon, but active bleeding was not observed, and the patient was followed up without additional hemostatic treatment in the remaining case. Early phase re-bleeding did not recur in the three cases. There was no case that required additional TAE or surgery (0%, 95% CI: 0–8%). No adverse events occurred in any of the patients (0%, 95% CI: 0–8%), and neither perforation nor abscess formation was observed (Table 2).

## 4. Discussion

Several hemostatic treatments for colonic diverticular hemorrhage, such as the endoscopic clipping method [6], epinephrine injection method [7,8], and coagulation method [7,9,10], have been performed. Ligation therapies, such as EBL [11,12], as well as endoscopic detachable snare ligation (EDSL) [13,14], are reportedly effective hemostatic treatments for colonic diverticular hemorrhage. A recent systematic review and meta-analysis of observational studies that evaluated and compared treatment outcomes among three groups, including the coagulation method, endoscopic clipping method, and ligation method, reported primary hemostasis 100% (95% CI: 95–100%), 99% (95% CI: 91–100%), and 99% (95% CI: 95–100%); early phase re-bleeding (within 30 days following primary hemostasis) 21% (95% CI: 1–51%), 19% (95% CI: 7–35%), and 9% (95% CI: 4–15%) (*p* = 0.332); additional TAE or surgery, 18% (95% CI: 0–61%), 8% (95% CI: 3–16%), and 0% (95% CI: 0–1%), respectively (*p* = 0.005). This indicated that ligation therapy was the most effective endoscopic treatment among the three groups [15]. In addition, another systematic review comparing the band ligation method and endoscopic clipping method indicated that no significant difference in the initial hemostasis rate was noted between the two groups. However, the pooled prevalence of early re-bleeding was significantly lower for band ligation than clipping (0.08 vs. 0.19, *p* = 0.012), and the pooled prevalence of the need for transcatheter arterial embolization or surgery was significantly lower for band ligation than clipping (0.01 vs. 0.02; heterogeneity test, *p* = 0.031) [16]. In our study, the OTSC method yielded a relatively better treatment outcome, as the procedure success rate was 100% (95% CI: 92–100%), early phase re-bleeding was 8.3% (95% CI: 8–22.5%), and additional TAE or surgery was 0% (95% CI: 0–8%), even though these results cannot be simply compared to the treatment outcome of the ligation method in those of previous reports [15,16].

The ligation method is considered to be a very efficient hemostatic procedure because of its high procedure success rate, low early-period re-bleeding rate, and low additional TAE or surgery rate [15,16,17]. However, there are two disadvantages of the ligation method. First, it may sometimes result in delayed perforation [18] or colonic diverticulitis [11,16] as adverse events, although the number reported is small. Second, the procedure is challenging because it requires adequate suction of the responsible diverticulum to the suction cap in cases involving a small diverticulum opening or a large diverticulum for the procedure to be successful. Ishii et al. reported that EBL could not be achieved in four (13%) out of 31 cases of colonic diverticular hemorrhage [19].

An OTSC is a shape-memorized clip made of nitinol. Although some reports have suggested the efficacy of OTSC as a hemostatic method for colonic diverticular bleeding, the number of cases was too small to validate this claim [20,21,22,23]. In this retrospective case series of the use of the OTSC method for colonic diverticular bleeding, the OTSC method was used in 36 cases of colonic diverticular bleeding. A feature of an OTSC is that it is not only closed, but “jumps” forward by ~4 mm when released. Owing to its “jumping effect,” adequate suction of a diverticulum to the suction cap is not required when using the OTSC method; however, it is definitely required when using the ligation method. Hemostasis can only be achieved if the colonic mucosa around the responsible diverticulum and the suction cap attached to the tip of the endoscope are in close contact in the OTSC method. Although in 47% of cases complete inversion of the diverticulum was not achieved, primary hemostasis using the OTSC method was achieved in all cases. Moreover, Yamazaki et al. reported a case in which the OTSC method enabled hemostasis after failure of EBL because of difficulties in adequate suction of the diverticulum. There were a few cases similar to the report in this study, and this suggests that the OTSC method enabled hemostasis, even in cases where endoscopic hemostasis would have been impossible because of the difficulty associated with adequate suction of the diverticulum by conventional EBL for colonic diverticular bleeding.

The adverse events of the OTSC method will now be discussed. Cases have been reported of an OTSC that led to esophageal perforation [24] or duodenal perforation [25] due to a tear in the walls of the digestive tract caused by the sharp tips of the OTSC when used for a purpose other than for colonic diverticular bleeding. However, when OTSC use is limited to colonic diverticular bleeding, there have been no reports of adverse events caused by the OTSC method, including this study [20,21,22,23]. It is believed that colonic perforation caused by an OTSC can be avoided if the diverticulum is sutured with the surrounding normal colonic mucosa, because the direct contact of the tip of an OTSC to the inner wall of the diverticulum that lacks a muscular layer could cause tears, resulting in perforations. Moreover, delayed perforation caused by the OTSC method was not observed in this study, and this may be attributable to two potential reasons. First, an OTSC itself has the ability to prevent colonic perforation, as it enables full-thickness resection at the central tissue mound of the inverted diverticulum [26]. Second, while all of the blood flow is blocked, including the responsible blood vessel and other blood vessels flowing into the diverticulum, as the mucosa is ligated circumferentially during conventional EBL, which can cause ischemia (Figure 4a,b), using an OTSC can preserve blood flow, and tends not to cause necrotic changes at the central mound, because an OTSC has spaces at the lateral sides that enable other blood vessels to flow into the diverticulum, thereby avoiding circumferential blood flow blockage (Figure 4a,c). In this study, the OTSC did not drop off from the responsible diverticulum in all three cases of re-bleeding, and the recurrence of active bleeding from the inverted diverticulum by OTSC was observed in two of the three cases, and this indicated that the OTSC did not cause complete blood flow blockage.

The OTSC method has some disadvantages. First, reinsertion of an endoscope is necessary after the diverticulum responsible for colonic diverticular bleeding is identified. The suction cap of an OTSC is long, and this makes the field of view narrow. Thus, identification of the bleeding diverticulum is challenging when using an OTSC system-mounted endoscope. Therefore, it is necessary to withdraw the endoscope when the responsible diverticulum is identified, mount the OTSC system at the tip of the endoscope, and re-insert it into the diverticulum, and this is cumbersome and time-consuming. Additionally, the OTSC system is very expensive.

The limitation of this study is that we examined a small number of cases (*n* = 36) for which the required number of cases was not statistically calculated, and the retrospective and observational nature of the study was conducted at a single facility. Multicenter prospective studies that evaluate the OTSC method in large cohorts are required in the future.

## 5. Conclusions

In conclusion, the OTSC method is effective and safe, and can be a novel hemostatic therapy for colonic diverticular bleeding.

## Figures and Tables

**Figure 1 jcm-10-02891-f001:**
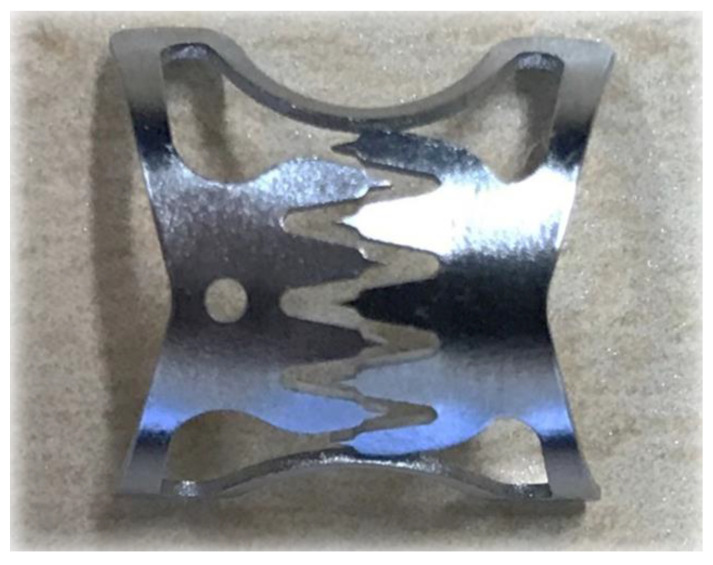
The structure of an over-the-scope clip (OTSC).

**Figure 2 jcm-10-02891-f002:**
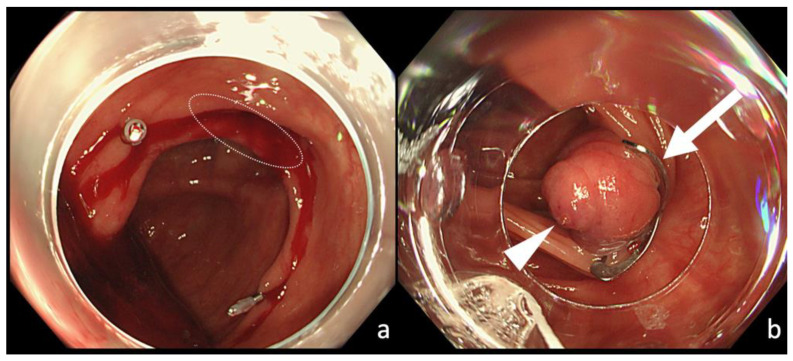
Hemostatic treatment using the OTSC method for colonic diverticular bleeding. (**a**) Place two clips near the bleeding diverticulum (white circle) for marking. (**b**) An OTSC (white arrow) released to the responsible diverticulum. The diverticulum is inverted and bulged by OTSC and then fixed. Hemostasis is achieved, and an exposed blood vessel (white arrowhead) is identified at the center mound.

**Figure 3 jcm-10-02891-f003:**
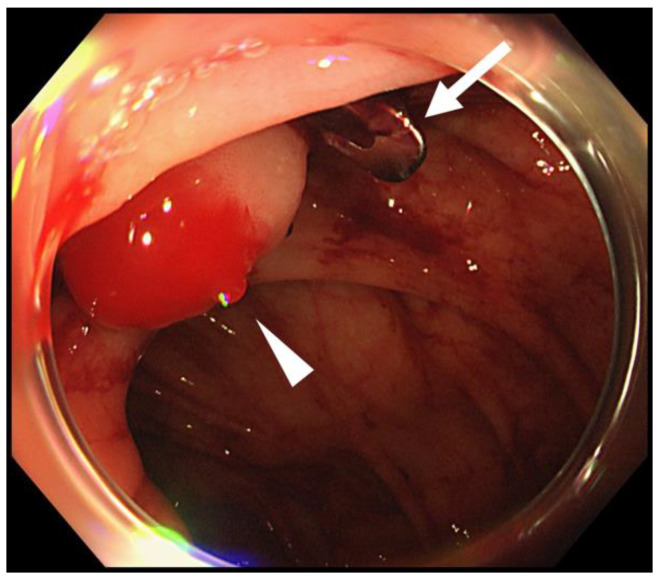
Colonoscopy performed when the early-period re-bleeding occurred after primary hemostasis by the OTSC method. Although an OTSC (white arrow) was still positioned and keeps the diverticulum inverted and bulged, active bleeding (white arrowhead) was observed from the top of the mound.

**Figure 4 jcm-10-02891-f004:**
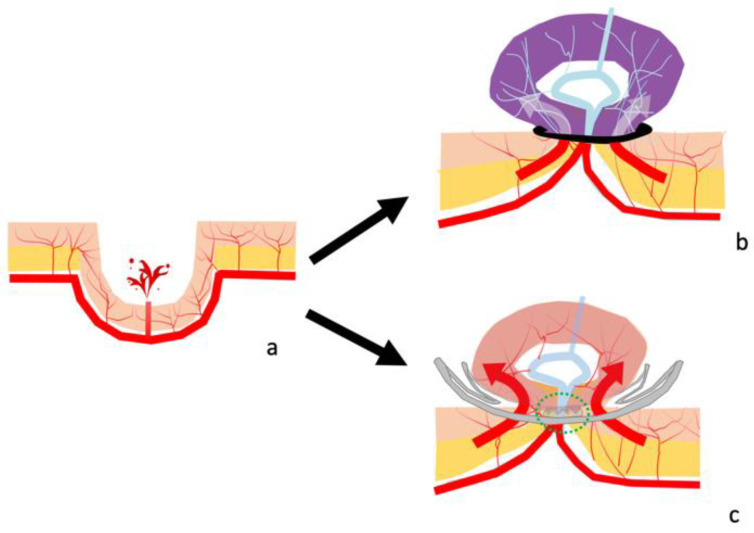
Schema of the endoscopic band ligation method (EBL) and over-the-scope clip method (OTSC). (**a**) Schema of colonic diverticular bleeding. (**b**) Schema after EBL. The diverticulum is ligated circumferentially by a band, and all of the blood vessels flowing into the inverted and bulged lesions are blocked (red and white arrow), causing ischemia. (**c**) Schema after an OTSC placement. The flow of the blood vessel captured by the OTSC is blocked (green circle), whereas the flow of other blood vessels through the space of the lateral sides of the OTSC to the colonic mucosa of the diverticulum remains (red arrow).

**Table 1 jcm-10-02891-t001:** Characteristics of patients treated by the OTSC method for colonic diverticular bleeding.

Factors	Total (*n* = 36)
Age (years), median (range)	78 (61–97)
Sex, male/female	25/11
History of previous treatment (EBL), *n*. (%)	2 (6)
Use of antiplatelet agents, *n*. (%)	14 (39)
Use of NSAIDs, *n*. (%)	6 (17)
Use of steroid, *n*. (%)	5 (14)
Shock index on admission, mean ± SD	0.68 ± 0.18
Hematocrit on admission, mean ± SD	30.1 ± 5.4
Units of PRBCs, mean ± SD	2.2 ± 3.8
Location in colon (C/A/T/D/S), *n*	4/18/2/4/8
Stigmata of hemorrhage (AB/NBVV), *n*	19/13
Complete inversion of a diverticulum, *n* (%)	19 (53)

OTSC: over-the-scope clip, SD: standard deviation, EBL: endoscopic band ligation, NSAID: nonsteroidal anti-inflammatory drug, PRBC: packed red blood cells, C: cecum, A: ascending colon, T: transverse colon, D: descending colon, S: sigmoid colon, AB: active bleeding, NBVV: non-bleeding visible vessel.

**Table 2 jcm-10-02891-t002:** Treatment outcomes of the OTSC method for colonic diverticular bleeding.

Outcomes	Total (*n* = 36)
Primary hemostasis achieved, *n* (%)	36 (100)
Early-period re-bleeding after OTSC, *n* (%)	3 (8)
Additional TAE or surgery, *n* (%)	0 (0)
Adverse event, *n* (%)	0 (0)

OTSC: over-the-scope clip, TAE: transcatheter arterial embolization.

## Data Availability

All the data used for this analysis can be confirmed at any time.

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
