# Peer review of "Efficacy of Over-The-Scope Clip Method as a Novel Hemostatic Therapy for Colonic Diverticular Bleeding"

_jcm, 2021, doi:10.3390/jcm10132891_

Round 1

Reviewer 1 Report

The study, despite being a retrospective study, is interesting and reaffirms the effectiveness and safety of the application of the OTSC method even in bleeding diverticular pathology.

However it would be useful to specify some details:

  1. why Traumatic and non-Atraumatic OTSCs were used?

    The same authors in the "discussion" specify that:

“because the direct contact of the tip of an OTSC to the inner wall of the diverticulum that lacks a muscular layer could cause tears resulting in perforation.”

Much lower risk of perforation if atraumatic OTSC clips are used (as generally recommended in the literature for haemostasis)

2. if in cases of active bleeding they first infiltrated with physiological and adrenaline to get a better view

3. Another useful information would be to know, if the data is available, what is the average time elapsed between the start of bleeding and the haemostasis procedure both in cases of first bleeding and in cases of "rescue therapy"

Author Response

Comments and Suggestions for Authors

The study, despite being a retrospective study, is interesting and reaffirms the effectiveness and safety of the application of the OTSC method even in bleeding diverticular pathology.

However it would be useful to specify some details:

  1. why Traumatic and non-Atraumatic OTSCs were used?

The same authors in the "discussion" specify that:

“because the direct contact of the tip of an OTSC to the inner wall of the diverticulum that lacks a muscular layer could cause tears resulting in perforation.”

Much lower risk of perforation if atraumatic OTSC clips are used (as generally recommended in the literature for hemostasis)

【Response】

Thank you for pointing out. Indeed, in a previous report we stated that “because the direct contact of the tip of an OTSC to the inner wall of the diverticulum that lacks a muscular layer could cause tears resulted in perforation.”

I added the following sentences to method64-66 of the main text.

Before suction, it is necessary to adjust the position of the endoscopic tip mounted with OTSC to the normal mucosa around the diverticulum, not to the inner wall of the di-verticulum lacking mucosal layer

Thank you for your very important point.

  1. if in cases of active bleeding they first infiltrated with physiological and adrenaline to get a better view

【Response】

Certainly, your suggesting adrenaline spray may provide a temporary hemostatic effect and improve the field of view of the endoscopic image. However, we did not use adrenaline spraying to improve visual field in this study.

  1. Another useful information would be to know, if the data is available, what is the average time elapsed between the start of bleeding and the haemostasis procedure both in cases of first bleeding and in cases of "rescue therapy"

【Response】

As you pointed out, the time required from the discovery of the bleeding diverticulum to the completion of hemostatic treatment is important as an index for evaluating the usefulness of treatment.

Unfortunately, this study was a retrospective study and could not be evaluated for treatment-related time. With reference to this result, next time we will proceed with a prospective research plan that includes treatment time as an evaluation item.

Reviewer 2 Report

This is a study on 36 colonic diverticular bleeding patients who were treated using the OTSC method. Although interesting, there are several limitations to be published as an original article.

  1. Due to the lack of control group, this study is closer to a case series rather than an original article.
  2. Study outcomes (procedure success rate, adverse events rate, early phase re-bleeding rate, and the requirement rate for additional treatment) should be compared between the case group (patients treated using the OTSC method) and the control group (patients treated using conventional method).
  3. There is no statistical evidence to prove that 36 patients are enough to prove the efficacy and safety of the OTSC method. Hence, adverse events rate and additional TAE or surgery rate were 0%. At least, sample size calculation should be performed.

Author Response

Comments and Suggestions for Authors

This is a study on 36 colonic diverticular bleeding patients who were treated using the OTSC method. Although interesting, there are several limitations to be published as an original article.

1.

Due to the lack of control group, this study is closer to a case series rather than an original article.

【Response】

Thank you for pointing out. You're right. However, this method is a new technology and has not been compared.

As you say, a positive comparison with existing technology is desired in the future, but in order to do so, we first summarized the results of this method and confirmed its safety.

2.

Study outcomes (procedure success rate, adverse events rate, early phase re-bleeding rate, and the requirement rate for additional treatment) should be compared between the case group (patients treated using the OTSC method) and the control group (patients treated using conventional method).

【Response】

Thank you for pointing out.

As you pointed out, it needs to be compared with existing methods.

However, since this is not a prospective comparative study, we are comparing the previously reported treatment results of existing treatments with the treatment results of this OTSC as shown below.

A recent systematic review and meta-analysis of observational studies that evaluated and compared treatment outcomes among three groups: coagulation method, endoscopic clipping method, and ligation method, reported primary hemostasis 100% (95%CI: 95–100%), 99% (95%CI: 91–100%), and 99% (95%CI: 95–100%); early phase re-bleeding (within 30 days following primary hemostasis) 21% (95%CI: 1–51%), 19% (95%CI: 7–35%), and 9% (95%CI: 4–15% ) (p=0.332); additional TAE or surgery, 18% (95%CI: 0–61%), 8% (95%CI: 3–16%), and 0% (95%CI: 0–1%), respectively (p=0.005). This indicated that ligation therapy was the most effective endoscopic treatment among the three groups [15]. In our study, the OTSC method yielded a relatively better treatment outcome as the procedure success rate was 100% (95%CI: 92–100%), early phase re-bleeding was 8.3% (95%CI: 8–22.5%), and additional TAE or surgery was 0% (95%CI: 0–8%), even though these results cannot be simply compared to those of previous reports.

However, as you pointed out, it should be compared and examined positively, so we will proceed with the research plan in the future with reference to this result.

Thank you for giving us the right direction.

3.

There is no statistical evidence to prove that 36 patients are enough to prove the efficacy and safety of the OTSC method. Hence, adverse events rate and additional TAE or surgery rate were 0%. At least, sample size calculation should be performed.

【Response】

Thank you for pointing out.

As you say, the number of cases requiring case study should be set statistically in a positive manner, but the sample size could not be set because this study was a retrospective study and did not have a control group.

This point is a very important limitation, so I added the following to the limitation part this time.

The limitation of this study is a small number of cases for which the required number of cases was not statistically calculated, and the retrospective observational nature of the study that was conducted at a single facility.

All your suggestions have greatly improved the content of this paper.

Thank you very much.

Round 2

Reviewer 2 Report

Thank you for submitting after revision.

Unfortunately, it is still a case series of 36 patients rather than an original article.

There is no sample size calculation that supports the study conclusions and findings. 

I am sorry that I cannot be more positive on this occasion.

Author Response

Thank you for the comments.

Later, we try to make RCT regarding this topics.